# Probabilistic Bisection Algorithm Provably Achieves Exponential Convergence

## Abstract

The probabilistic bisection algorithm (PBA) extends the classical binary search to settings with noisy responses, and is a foundational algorithm commonly used in basic problems such as root-finding. Despite its strong empirical success, its theoretical property, particularly the convergence rate, remains unclear. This paper establishes that PBA converges at a geometric rate, providing a rigorous justification for its empirical efficiency. Notably, this rate is optimal in the sense that it matches the performance of classical binary search under noiseless responses. The core of our analysis lies in directly characterizing the dynamics of PBA queries, which had not been examined in the prior literature. We show that the queries oscillate around the truth but steadily draw closer, thus leading to an estimator that rapidly concentrates on the truth. Beyond resolving the long-standing question of PBA's convergence, our developed techniques offer new tools for analyzing PBA's dynamics, which may be of independent interest.

## 1 Introduction

Binary search is a fundamental algorithm that addresses the core challenge of **efficiently locating a target within an ordered space** using the principle of divide-and-conquer. It underpins a wide range of modern algorithms in computer science, statistics, and applied mathematics (Knuth, 1997; Karp & Kleinberg, 2007; Waeber et al., 2013), and serves as a building block for systems and methods from multidimensional data to search on graphs and trees (Bentley, 1975; Nowak, 2009; Emamjomeh-Zadeh et al., 2016; Rodriguez & Ludkovski, 2020a). Classical applications include fast key retrieval in large databases and numerical root-finding in engineering and economics. For instance, consider finding a unique root of a monotone function $h : [0, 1] \to \mathbb{R}$ where one can query only the sign of $h(x)$. When the response to each query $x$ is noiseless, a binary search algorithm efficiently locates the root by halving the search interval each round. After $n$ queries, the remaining interval has length $2^{-n}$, achieving the optimal exponential convergence rate.

In practice, however, the observed responses are often noisy, e.g., due to transmission and measurement error, meaning that they have a chance to be incorrect. Motivated by noisy channel coding, the Probabilistic Bisection Algorithm (PBA, Horstein, 1963) extends binary search to handle noisy labels. Compared to binary search, PBA adopts a Bayesian approach to select the query. In the 1-D root-finding setup, PBA maintains a probability distribution with density $f_t$ over the support $[0, 1]$, representing the likelihood of each point being the true root. At each round $t$, PBA queries the median $x_t$ of this distribution and receives a noisy response $y_t$ indicating the sign at $x_t$. The belief is then updated via Bayes' rule given $y_t$. For instance, if $y_t$ is positive, then $f_t(x) = 2(1 - p)f_{t-1}(x)$ for $x \leq x_t$ and $f_t(x) = 2pf_{t-1}(x)$ for $x > x_t$, where $p$ is the noise level. The process repeats until termination, with the final estimator being the last query.

Despite strong empirical performance, (Waeber, 2013; Frazier et al., 2019; Rodriguez & Ludkovski, 2020a;b), PBA's theoretical property, particularly its convergence rate, remains poorly understood. The difficulty stems from its **intricate query process over a continuous search space**. In comparison, the so-called noisy binary search typically focuses on a finite search set and enjoys well-understood guarantees. It has been shown that locating a target

among $H$ elements with error probability at most $\delta$ requires only $O(\log(H/\delta))$ queries (Karp & Kleinberg, 2007; Nowak, 2009; Emamjomeh-Zadeh et al., 2016). However, these results rely crucially on the discretized structure of the search space *such as searching nodes on a path (Aslam & Dhagat, 1991; Karp & Kleinberg, 2007) or a graph (Emamjomeh-Zadeh et al., 2016; Dereniowski et al., 2019).* In contrast, PBA addresses a continuous domain with uncountably many possible queries, which requires fundamentally different analysis tools.

Historically, analyzing PBA's performance has been difficult due to the continuous nature of the query sequence, which makes tracking the estimation non-trivial. As a result, prior efforts either (1) adopted a discretized version of PBA, or (2) invoked a Bayesian framework where the unknown truth is modeled as a continuous random variable. However, these approaches are unable to directly characterize the convergence behavior of the original PBA given a fixed truth. Specifically, Burnashev & Zigangirov (1974) proposed a discretized version of PBA, which we refer to as the BZ algorithm. BZ restricts queries to a finite grid $\{0, 1/K, 2/K, \ldots, 1\}$ for some constant $K$. With carefully modified update and query rules, they proved that BZ attains exponential convergence when $K$ adapts to the query size (Burnashev & Zigangirov, 1974; Castro & Nowak, 2008). However, a pre-selected and fixed $K$ is required to run BZ in practice, thus such a convergence rate cannot be expected. Waeber et al. (2013) analyzed PBA in a Bayesian setting. By modeling the root as a random variable $X^*$ uniformly distributed on $[0, 1]$, they proved that $\mathbb{E}|X^* - \widehat{X}_n|$ decays geometrically, where $\widehat{X}_n$ is the PBA estimate after $n$ queries. However, this result hinges critically on the assumption that $X^*$ is a continuous random variable. Therefore, this analysis does not apply to real-world tasks where the ground truth is a fixed but unknown constant, such as root-finding and boundary detection problems.

A closer inspection of these approaches shows that the main barrier to analyzing the original PBA, again, lies in the complex, location-dependent behavior of its queries. Both the discretized analysis of Burnashev & Zigangirov (1974) and the Bayesian analysis of Waeber et al. (2013) exploit a simplifying property: at every round a quantity that upper-bounds the estimation error is expected to decrease, regardless of where the queries fall. Unfortunately, this guarantee breaks down when analyzing the original PBA with a fixed ground truth, as the improvement in accuracy depends delicately on the query locations (with further discussion in Subsection 2.2).

Our work demonstrates that understanding the query behavior of PBA is both essential and powerful in tackling this problem. In Subsection 2.3, we develop new analytical techniques that measure the improvement contributed by the query at each round, and characterize the number of queries that lead to a better estimation, an aspect not studied in the prior literature. These tools allow us to directly study the dynamics of PBA queries. Intuitively, we show that the queries oscillate around the ground truth but steadily draw closer, driving the posterior distribution to concentrate sharply at the ground truth.

Building on these tools, we prove that **PBA converges at an exponential rate for any fixed, unknown ground truth.** The rate we establish is optimal, matching the geometric convergence achievable by classical binary search with noiseless feedback. This result settles the long-standing theoretical question of whether PBA retains its empirical efficiency under noisy responses (Waeber et al., 2013). Moreover, our developed tools provide a fine-grained understanding of PBA's query process, which may be of independent interest for other adaptive algorithms.

The rest of the paper is organized as follows. Section 2 present our main result, the convergence rate of PBA for one-dimensional data. Simulation experiments are conducted in Appendix D, and we discuss the extension to the high-dimensional data in Appendix C. We conclude the paper with further discussions in Section 3.

## 2 CONVERGENCE RATE OF PBA

### 2.1 SETUP

We cast the root-finding problem as a special case of binary classification. Consider a learner seeking to identify the unknown classifier $h_{\theta^*}(x) = \mathbb{1}_{x \geq \theta^*}$ within a hypothesis class $\mathcal{H} = \{h_\theta : \theta \in [0,1]\}$, where $\mathbb{1}_{(\cdot)}$ is an indicator function. Let $p \in (0, 1/2)$ denote the noise level in the response. In this formulation, $\theta^*$ is the unknown root, and each response is flipped independently with probability $p$. Specifically, for any query $X$, the observed response $Y$ satisfies that $\mathbb{P}(Y = h_{\theta^*}(X)) = 1 - p$ and $\mathbb{P}(Y = 1 - h_{\theta^*}(X)) = p$. We note that our results also extend to the more general setting where $\mathbb{P}(Y = 1 - h_{\theta^*}(X)) \leq p$, as elaborated in Appendix B.

**Probabilistic Bisection Algorithm (PBA).** A learner can use PBA to efficiently estimate $\theta^*$ as follows. Let $P_0$ be a uniform prior distribution such that its density function is $f_0(x) = 1, x \in [0,1]$. At round $i \geq 1$, PBA will select a query $X_i$ as the median of $\mathbb{P}_{i-1}$, i.e.,

$$\mathbb{P}_{i-1}(X \leq X_i) = 1/2.$$

After observing the corresponding label $Y_i$, PBS updates the posterior distribution as follows:

(1) If $Y_i = 1$, $f_i(x) = \begin{cases} 2(1-p)f_{i-1}(x), & x \leq X_i, \\ 2pf_{i-1}(x), & x > X_i, \end{cases}$

(2) If $Y_i = 0$, $f_i(x) = \begin{cases} 2pf_{i-1}(x), & x \leq X_i, \\ 2(1-p)f_{i-1}(x), & x > X_i. \end{cases}$

The posterior distribution at round $i$ is $\mathbb{P}_i(t) = \int_0^t f_i(x)dx$. The final estimator of $\theta^*$ after $n$ rounds is $\widehat{\theta}_n := X_{n+1}$.

*Remark* 1 (Prior and Posterior Distributions). In our setting the unknown root $\theta^*$ is fixed. The distribution $\mathbb{P}_i$ represents the learner's belief about $\theta^*$: at round $i$, they believe that the probability that $\theta^* \leq t$ is given by $\mathbb{P}_i(t)$. We use the terms *prior* and *posterior* in keeping with the PBA literature, where the algorithm is commonly interpreted from a Bayesian perspective.

### 2.2 EXPONENTIAL CONVERGENCE RATE

Throughout the paper, $c$ and $C$ are either universal constants or constant of $p$ only, though their value may vary from line to line. We use the terms 'root', 'truth', and 'ground truth' interchangeably. The complete proof of Theorem 1 is included in Appendix A.

**Theorem 1** (Exponential Convergence Rate of PBA)**.** *For the PBA estimator $\widehat{\theta}_n$, we have*

$$\mathbb{E}|\widehat{\theta}_n - \theta^*| \leq 3e^{-Cn},$$

*where $C > 0$ is a constant of $p$ only.*

**Key Challenges and Contributions.** We assume that $\theta^* \in (0, 1)$ for illustration purpose. The basic idea is to show that $\widehat{\theta}_n$ lies within a small interval around $\theta^*$ with high probability. Partition $[0,1]$ into $K$ intervals $[(i-1)/K, i/K)$, $i = 1, 2, \ldots, K$. Then there exists some $i^*$ such that $\theta^* \in [\delta_{i^*-1}, \delta_{i^*})$. We can prove that

$$\mathbb{P}\big(\widehat{\theta}_n \in [\delta_{i^*-1}, \delta_{i^*})\big) \geq 1 - 2K^2(K+1)e^{-Cn}. \tag{1}$$

Choosing $K = e^{Cn/4}$ yields the desired result.

Eq. 1 is equivalent to showing that the PBA estimator is unlikely to be much larger or smaller than $\theta^*$. By symmetry, it suffices to prove the upper-tail bound:

$$\mathbb{P}\big(\widehat{\theta}_n > \delta_{i^*}\big) \leq K^2(K+1)e^{-Cn}. \tag{2}$$

The key challenge is to establish the exponential decay result in Eq. 2.

We emphasize that although similar bounds were obtained in (Burnashev & Zigangirov, 1974; Waeber et al., 2013), their proofs rely on the argument that

$$\mathbb{E}(M_{i+1} - M_i \mid M_i) \leq -C, \tag{3}$$

where $M_i$ is a quantity, in particular, $\log(M_\theta(i))$ in Hero et al. (2007, Theorem 8.1) and $\log(A_i \wedge (1 - A_i))$ in Waeber et al. (2013, Proposition 5.3), that can upper bound $\mathbb{P}(\widehat{\theta}_i > \delta_{i^*})$. That is, under the discretization or the Bayesian setting, there exists a stochastic process $M_i$'s that is equivalent to a geometric random walk with negative shift. Hence, Eq. 3 guarantees that the estimator moves closer to the truth after each query (in expectation), regardless of query location. However, our analysis reveals that this property fails for the original PBA: **the accuracy improvement depends critically on the query position, and improvement is not always guaranteed.**

To overcome this challenge, we conduct a finer-grid analysis of PBA's query dynamics, as detailed in the next subsection. Consequently, our proof of Eq. 2 employs a fundamentally different argument from those in (Burnashev & Zigangirov, 1974; Waeber et al., 2013), which constitutes a key methodological contribution of this work.

### 2.3 Analysis of Query Behaviors

This subsection introduces two novel propositions on query behaviors, which are key for deriving Eq. 2. First, we recall a key property of PBA query: it is the median of posterior belief, meaning that $X_{n+1}$ satisfies

$$\mathbb{P}_n(X_{n+1}) = 1/2. \tag{4}$$

This equation establishes a direct connection between the query location and the posterior probability mass over intervals, which will play a critical role in our analysis.

We introduce the following critical quantities before presenting the results. Let $\delta$ be a constant such that $\theta^* < \delta < 1$. We divide the interval $[0, 1]$ into three sub-intervals:

$$I_1 := [0, \theta^*], I_2 := (\theta^*, \delta), I_3 := [\delta, 1],$$

and define

$$a_i^{(j)}(\delta) := \mathbb{P}_i(X \in I_j) := \int_{I_j} f_i(x)dx.$$

We will omit the dependence on $\delta$ when clear from context. Namely, $a_i^{(j)}$ is the posterior probability that the estimator $\widehat{\theta}_{i+1}$ lies in the $j$-th sub-interval after the $i$-th query.

*Remark* 2 (Motivation for $a_i^{(j)}$). At each round $i$, the query must fall into one of three sub-intervals, which is completely determined by $a_{i-1}^{(j)}$'s. Recall that $\widehat{\theta}_n = X_{n+1}$. By Eq. 4, a large estimator $X_{n+1} > \delta$ implies $a_n^{(3)} = \int_\delta^1 f_n(x)dx > 1/2$. Thus, to establish that the probability of such a large estimator is exponentially small, it suffices to show that $a_n^{(3)}$ is unlikely to become large. It turns out that understanding the behavior of PBA queries is necessary for deriving such a result, which further relies on tracking the change of all three posterior probabilities.

For any realization of $X_i, Y_i$'s, we further define

$$N_j := \sum_{i=1}^n \mathbb{1}_{X_i \in I_j}, G_j := \{i \in [1, n] : X_i \in I_j\}, j = 1, 2, 3.$$

$N_j$ is the number of total occurrences of the event $X_i \in I_j$, and $G_j$ contains the corresponding indices. We also define the following stopping times:

$$\tau_0 = 0, \ \tau_i = \inf\left\{t : t > \tau_{i-1}, \text{sign}(a_t^{(1)} - 1/2) \neq \text{sign}(a_{t-1}^{(1)} - 1/2)\right\}, i = 1, 2, \ldots,$$

where $\text{sign}(x) = 1$ for $x \geq 0$ otherwise $\text{sign}(x) = -1$. Namely, $\tau_i$ is the $i$-th time such that $a_t^{(1)}$ across $1/2$, meaning that the query's location transits from $I_1$ to $I_2 \cup I_3$ or vice versa. Finally, we define

$$T := \sup\{i : i \geq 0, \tau_i \leq n\}, \tag{5}$$

which is the number of total times that the query crosses the truth $\theta^*$.

**Proposition 1.** *Let $M_i(\delta) := a_i^{(3)}(\delta)/a_i^{(2)}(\delta)$. For some constant $C_1, C_2 > 0$ of $p$ only, we have*

$$\mathbb{P}(M_n/M_0 \leq e^{-C_1 n}) \geq 1 - e^{-C_2 n},$$

*which implies $\mathbb{E}M_n \leq e^{-C_3 n}/|\theta^* - \delta|$ for some positive constant $C_3$.*

**Proposition 2.** *There exists a constant $\eta$ only depending on $p$, such that $\mathbb{E}(T) \geq \eta n$.*

**Implications.** Together, Propositions 1 and 2 yield a sharp picture of the dynamics: the posterior distribution of PBA rapidly concentrates around $\theta^*$, while the queries themselves are expected to oscillate across the truth. In other words, the queries repeatedly swing around $\theta^*$ but with steadily shrinking amplitude, driving convergence. This insight provides a fundamental explanation for the empirical success of PBA.

Regarding the convergence rate, Proposition 1 shows that $M_n$ decays exponentially fast, which immediately implies an exponentially decaying probability of a large estimator. To see it, Markov's inequality gives that

$$\mathbb{P}(a_n^{(3)} \geq \epsilon) \leq \mathbb{P}(M_n \geq \epsilon) \leq \frac{\mathbb{E}M_n}{\epsilon} \leq \frac{e^{-Cn}}{|\theta^* - \delta|\epsilon}.$$

As a result, $\mathbb{P}(X_{n+1} \geq \delta) \leq \mathbb{P}(a_n^{(3)} \geq 1/2) \leq 2e^{-Cn}/|\theta^* - \delta|$, establishing the kep step (2) in Theorem 1.

**Proof Sketch.** The core idea behind the proof of Proposition 1 is to show that $\ln(M_i)$'s form a supermartingale, which decreases when the query lies in $I_2$ or $I_3$. We note that $\ln(M_i)$ remains unchanged when $X_i \in I_1$, and the decrease can be arbitrarily small when $X_i \in I_2$. Fortunately, we find that $\ln(M_i)$ decreases by at least a constant amount when the query crosses the truth. That is, when $X_{i-1} < \theta^* \leq X_i$ or $X_{i-1} \geq \theta^* > X_i$. This boundary-crossing behavior is characterized by $T$. Hence, to ensure that $\ln(M_n)$ becomes sufficiently small, it suffices to show that $\mathbb{E}(T)$ grows linearly with $n$, as established in Proposition 2.

Technically, Propositions 1 and 2 hinge on a careful analysis of the changes in $a_i^{(j)}$ and their combinations such as $M_i$. These changes depend on the query location and leads to a discussion of three cases: $X_i \in I_1$, $X_i \in I_2$, and $X_i \in I_3$. To prove Proposition 2, first we need to construct appropriate sub- or super-martingales from the posterior probabilities. We then show that the queries cross the truth sufficiently often by analyzing the boundary-crossing times $\tau_i$ and invoking the stopping time theorem. This, together with a martingale concentration inequality, ensures a significant reduction in $M_i$, thereby completing the proof of Proposition 1. The full details of these two propositions are presented below.

*Remark* 3 (Motivation of $M_i$). Proposition 1 focuses on analyzing $M_n$ rather than $a_n^{(3)}$. The quantity $M_n$ is deliberately and carefully designed, not an arbitrary combination of the $a_n^{(j)}$'s. The key reason is that the evolution of $a_n^{(j)}$ depends intricately on the query locations, making them difficult to control directly. To establish Eq. 2, we seek a quantity that is guaranteed to be monotone on average across rounds. However, the $a_n^{(j)}$'s alone do not exhibit this property for all possible query positions. By introducing a ratio-based structure, $M_n$ (specifically, its logarithm) acquires this desirable monotonicity, enabling a tractable analysis.

**Proof of Proposition 1.**

*Proof.* We prove this result by three steps: (1) $M_i$ is expected to decrease or maintain the same at each round, (2) there is a sufficient number of time steps such that $M_i$ is expected to decrease, (3) evoking a concentration inequality.

**Step 1:** Depending on the position of $X_i$, we discuss the update of $M_i$ in three cases as follows.

**Case 1, $X_i \in I_1$.** Clearly, by the update rule of PBS, $a^{(2)}$ and $a^{(3)}$ will be multiplied by $2(1-p)$ (when $Y_i = 0$) or $2p$ (when $Y_i = 1$) simultaneously. As a result, $M_i = M_{i-1}$ in this case.

**Case 2, $X_i \in I_2$.** Now, a correct label ($Y_i = 1$ with probability $1 - p$) leads to $a_i^{(1)} = 2(1-p)a_{i-1}^{(1)}$ and $a_i^{(3)} = 2pa_{i-1}^{(3)}$. As a result,

$$M_i/M_{i-1} = \frac{2pa_{i-1}^{(2)}}{1 - 2(1-p)a_{i-1}^{(1)} - 2pa_{i-1}^{(3)}} \in \left(\frac{p}{1-p}, 1\right).$$

A wrong label leads to

$$M_i/M_{i-1} = \frac{2(1-p)a_{i-1}^{(2)}}{1 - 2pa_{i-1}^{(1)} - 2(1-p)a_{i-1}^{(3)}} \in \left(1, \frac{1-p}{p}\right).$$

For notation simplicity, we denote $q_1 = a_{i-1}^{(1)}, q_2 = a_{i-1}^{(2)}, q_3 = a_{i-1}^{(3)}$. Some important properties of them are summarized in Lemma 1. Evoking Lemma 1, we have

$$\mathbb{E}\left(\frac{M_i}{M_{i-1}}\right) = (1-p)\frac{2pq_2}{1 - 2(1-p)q_1 - 2pq_3} + p\frac{2(1-p)q_2}{1 - 2pq_1 - 2(1-p)q_3}$$

$$= \frac{2p(1-p)q_2}{q_2 - (1-2p)(q_1 - q_3)} + \frac{2p(1-p)q_2}{q_2 + (1-2p)(q_1 - q_3)}$$

$$= \frac{4p(1-p)(q_2)^2}{(q_2)^2 - (1-2p)^2(q_1 - q_3)^2}$$

$$= 1 - \frac{(1-2p)^2\{(q_2)^2 - (q_1 - q_3)^2\}}{(q_2)^2 - (1-2p)^2(q_1 - q_3)^2}$$

$$< 1.$$

The last step is due to $q_2 > |q_1 - q_3|$ and the positivity of denominator.

Moreover, for any $\epsilon \in (0, 1/2)$, when $q_1, q_3 \le (1 - \epsilon)/2$, the fourth point of Lemma 1 gives that $q_2 - |q_1 - q_3| \ge \epsilon$ and

$$\mathbb{E}\left(\frac{M_i}{M_{i-1}}\right) = 1 - \frac{(1-2p)^2\{(q_2)^2 - (q_1 - q_3)^2\}}{(q_2)^2 - (1-2p)^2(q_1 - q_3)^2}$$

$$\le 1 - (1-2p)^2\epsilon^2.$$

As a result, Jensen's Inequality gives

$$\mathbb{E}\left\{\ln\left(\frac{M_i}{M_{i-1}}\right)\right\} \le \ln(1 - (1-2p)^2\epsilon^2) \le -(1-2p)^2\epsilon^2.$$

**Case 3, $X_i \in I_3$.** In this case, a correct label ($Y_i = 1$ with probability $1 - p$) leads to $a_i^{(2)} = 2(1-p)a_{i-1}^{(2)}$ and $1 - a_i^{(3)} = 2(1-p)(1 - a_{i-1}^{(3)})$ , so that

$$M_i/M_{i-1} = \frac{1 - 2(1-p)(1 - a_{i-1}^{(3)})}{2(1-p)a_{i-1}^{(3)}} \in \left(\frac{p}{1-p}, \frac{1}{2(1-p)}\right).$$

Similarly, a wrong label results in

$$M_i/M_{i-1} = \frac{1 - 2p(1 - a_{i-1}^{(3)})}{2pa_{i-1}^{(3)}} \in \left(\frac{1}{2p}, \frac{1-p}{p}\right).$$

As a result, we have

$$\mathbb{E}\left\{\ln\left(\frac{M_i}{M_{i-1}}\right)\right\} = (1-p)\ln\left(\frac{1 - 2(1-p)(1 - a_{i-1}^{(3)})}{2(1-p)a_{i-1}^{(3)}}\right) + p\ln\left(\frac{1 - 2p(1 - a_{i-1}^{(3)})}{2pa_{i-1}^{(3)}}\right).$$

Let

$$h(x) := (1-p)\ln\left(\frac{1 - 2(1-p)(1 - x)}{x}\right) + p\ln\left(\frac{1 - 2p(1 - x)}{x}\right).$$

Its first derivative is

$$h'(x) = \frac{2(1-p)^2}{1 - 2(1-p)(1-x)} + \frac{2p^2}{1 - 2p(1-x)} - \frac{1}{x}.$$

We have

$$\begin{aligned}
h'(x) > 0 &\iff 2(1-p)^2 x + 2p^2 x > \{1 - 2(1-p)(1-x)\}\{1 - 2p(1-x)\} \\
&\iff (2 - 4p + 4p^2)x > 1 - 2(1-x) + 4p(1-p)(1-x) \\
&\iff 2x > 2x - 1 + 4p(1-p) \\
&\iff 1 > 4p(1-p),
\end{aligned}$$

which is true since $p \in (0, 1/2)$. Since $\mathbb{E}\left\{\ln\left(\frac{M_i}{M_{i-1}}\right)\right\} = h(a_{i-1}^{(3)}) - \ln(2) + H(p)$ where $H(p) = -p\ln(p) - (1-p)\ln(1-p)$ is the binary entropy function, we know that $\mathbb{E}\left\{\ln\left(\frac{M_i}{M_{i-1}}\right)\right\}$ achieves its maximum when $a_{i-1}^{(3)} = 1$, leading to

$$\mathbb{E}\left\{\ln\left(\frac{M_i}{M_{i-1}}\right)\right\} \leq -\ln(2) + H(p) < 0.$$

**Step 2:** We show that $M_i$ is expected to strictly decrease for sufficient number of rounds. Specifically, we know that $\mathbb{E}\left\{\ln\left(\frac{M_i}{M_{i-1}}\right)\right\}$ is strictly smaller than zero if (1) $X_i \in I_2$, and $q_1, q_3 < (1-\epsilon)/2$; and (2) $X_i \in I_3$. For a given realization of $X_i, Y_i$'s, the latter case happens for $N_3$ times. We define the number of the first case as

$$N_2'(\epsilon) := |G_2'(\epsilon)|, \quad G_2'(\epsilon) := \{i : X_i \in I_2, q_1, q_3 \leq (1-\epsilon)/2\}.$$

We will omit $\epsilon$ in the following as it will be chosen as a constant of $p$ solely.

Next, we show that $N_2' + N_3 \geq \eta' n$ with high prob for some $\eta'$. The idea is to show that each down-crossing of $\tau_i$ leads to an instance of $G_2'$ or $G_3$ with a constant probability, and Proposition 2 shows that such down-crossing happens sufficiently often. Let us consider each time $a_t^{(1)}$ goes down and crosses $1/2$. Suppose $a_{t-1}^{(1)} > 1/2$ and $a_t^{(1)} \leq 1/2$. By update rule, we have $2p \leq a_t^{(1)} \leq 1/2$, thus either $X_{t+1} \in I_2$ or $X_{t+1} \in I_3$.

(Step 2.1) We have $t + 1 \in G_2' \cup G_3$ when $X_{t+1} \in I_3$ or $X_{t+1} \in I_2$ with $a_t^{(1)}, a_t^{(3)} \leq (1-\epsilon)/2$.

(Step 2.2) Now, suppose $t \notin G_2' \cup G_3$, namely $X_{t+1} \in I_2$ and at least one of $a_t^{(1)}, a_t^{(3)}$ is larger than $(1-\epsilon)/2$.

We first consider the case where $a_t^{(1)} > (1-\epsilon)/2$. With probability $p$, $Y_{t+1}$ is a wrong label, and $X_{t+2} \in I_1$ since $a_{t+1}^{(1)} = 2(1-p)a_t^{(1)} > 1/2$ for any $\epsilon \in (0, 1 - 1/(2 - 2p))$. With probability $1 - p$, $Y_{t+1}$ is a correct label, leading to (i) $X_{t+2} \in I_3$, or (ii) $X_{t+2} \in I_2$. While (i) automatically leads to $t + 2 \in G_3$, (ii) again leads to two possible outcomes: (ii.a) $X_{t+3} \in I_3$, or (ii.b) $X_{t+3} \in I_2$. We note that (ii.b) results in $t + 3 \in G_2'$ for a sufficiently small $\epsilon$. To see it, we have $a_{t+2}^{(1)} = 4p(1-p)a_t^{(1)} < (1-\epsilon)/2$ and $a_{t+2}^{(3)} = 4p(1-p)a_t^{(3)} < (1-\epsilon)/2$ for any $\epsilon \in (0, (1-2p)^2)$. Next, we consider the case $a_t^{(3)} > (1-\epsilon)/2$. With probability $p$, $Y_{t+1}$ is a wrong label, we therefore have $t + 2 \in N_3$ because $a_{t+1}^{(3)} = 1 - 2p(1 - a_t^{(3)}) > 1 - p(1+\epsilon) > 1/2$ for any $\epsilon \in (0, (2p)^{-1} - 1)$. Combining these two cases, we have that with probability at least $p$, such time step $t$ will lead to an occurrence of $N_2'$ or $N_3$ before the next occurrence of $a_t^{(1)}$ going down and crossing $1/2$.

(Step 2.3) WLOG, let $a_0^{(1)} < 1/2$ as explained in the proof of Proposition 2. Let $R_k$ denoting whether $a_{\tau_{2k}}^{(1)}$ leads to an occurrence of $N_2'$ or $N_3$, we have $R_k$ being IID Bernoulli random variables with $\mathbb{P}(R_k = 1) \geq p$. We can therefore construct a sub-martingale

$S_l = \sum_{k=1}^{l} R_k - pl$, $l = 1, 2, \ldots$, and $S_0 = 0$. Now, applying the optional stopping theorem, we have $\mathbb{E}S_{\lfloor T/2 \rfloor} \geq \mathbb{E}S_0 = 0$, yielding

$$\mathbb{E}(N_2' + N_3) \geq \mathbb{E}\left(\sum_{k=1}^{\lfloor T/2 \rfloor} R_k\right) \geq p\mathbb{E}(\lfloor T/2 \rfloor).$$

Now, evoking Proposition 2, we have $\mathbb{E}(N_2' + N_3) \geq \eta' n$ for $\eta' = p\eta/2$.

**Step 3:** Finally, we show that $M_n$ is small with high probability by applying Azuma-Hoeffding inequality. Note that

$$M_n = M_0 \exp\left\{\sum_{i=1}^{n} \ln(M_i/M_{i-1})\right\}.$$

Step 1 indicates that $\sum_{i=1}^{n} \ln(M_i/M_{i-1})$ is a super-martingale with respect to $n$, because

$$\mathbb{E}\left\{\sum_{i=1}^{n} \ln(M_i/M_{i-1}) \mid \sum_{i=1}^{n-1} \ln(M_i/M_{i-1})\right\} = \mathbb{E}\ln(M_n/M_{n-1}) \leq 0.$$

Moreover, all $\ln(M_i/M_{i-1})$'s have a uniform upper bound on their absolute value and variance, denoted as $B_1, B_2 > 0$, respectively. Let $C_6 := \min\{(1-2p)^2\epsilon^2, \ln(2) - H(p)\} > 0$ and $\zeta = \eta' C_6/2$. Azuma-Hoeffding's inequality gives that

$$\mathbb{P}\left(\sum_{i=1}^{n} \ln(M_i/M_{i-1}) > \mathbb{E}\left(\sum_{i=1}^{n} \ln(M_i/M_{i-1})\right) + n\zeta\right) \leq e^{-2n\zeta^2}$$

$$\Longleftrightarrow \mathbb{P}\left(\sum_{i=1}^{n} \ln(M_i/M_{i-1}) > -\mathbb{E}(N_2' + N_3)C_6 + n\zeta\right) \leq e^{-2n\zeta^2}$$

$$\Longleftrightarrow \mathbb{P}\left(\sum_{i=1}^{n} \ln(M_i/M_{i-1}) > -\eta' C_6 n/2\right) \leq e^{-2n\zeta^2}$$

As a result, with probability at least $1 - e^{-2n\zeta^2}$, we have

$$M_n \leq M_0 \exp(-n\eta' C_6/2).$$

We therefore complete the proof by noting $M_0 = (1-\delta)/(\delta - \theta^*)$ since $f_0(x) = 1$ for all $x \in [0,1]$. $\square$

**Proof of Proposition 2.**

*Proof.* Let $b_i^{(1)} = 1 - a_i^{(1)}$. Depending on the position of $X_i$, The change of $a_i^{(1)}$ in each round is also categorized into three cases.

**Case 1,** $X_i \leq \theta^*$. A correct label ($Y = 0$ with probability $1 - p$) leads to $1 - a_i^{(1)} = 2(1-p)(1 - a_{i-1}^{(1)})$. Therefore,

$$\frac{a_i^{(1)}}{a_{i-1}^{(1)}} = \frac{1 - 2(1-p)(1 - a_{i-1}^{(1)})}{a_{i-1}^{(1)}}, \quad \frac{b_i^{(1)}}{b_{i-1}^{(1)}} = 2(1-p),$$

A wrong label leads to

$$\frac{a_i^{(1)}}{a_{i-1}^{(1)}} = \frac{1 - 2p(1 - a_{i-1}^{(1)})}{a_{i-1}^{(1)}}, \quad \frac{b_i^{(1)}}{b_{i-1}^{(1)}} = 2p.$$

Therefore,

$$\mathbb{E}\left\{\ln\left(\frac{a_i^{(1)}}{a_{i-1}^{(1)}}\right)\right\} = (1-p)\ln\left(\frac{1 - 2(1-p)(1 - a_{i-1}^{(1)})}{a_{i-1}^{(1)}}\right) + p\ln\left(\frac{1 - 2p(1 - a_{i-1}^{(1)})}{a_{i-1}^{(1)}}\right) < 0.$$

The last inequality is because function $h(a_{i-1}^{(1)}) := (1-p)\ln\left(\frac{1-2(1-p)(1-a_{i-1}^{(1)})}{a_{i-1}^{(1)}}\right) + p\ln\left(\frac{1-2p(1-a_{i-1}^{(1)})}{a_{i-1}^{(1)}}\right)$ monotonously increases when $a_{i-1}^{(1)} \in (1/2, 1)$, which can be verified by taking its first derivative.

Also,

$$\mathbb{E}\left\{\ln\left(\frac{b_i^{(1)}}{b_{i-1}^{(1)}}\right)\right\} = (1-p)\ln(2(1-p)) + p\ln(2p) = \ln(2) - H(p) > 0,$$

where $H(p) = -p\ln(p) - (1-p)\ln(1-p)$ is the binary entropy function.

**Case 2, $X_i \geq \delta$.** A correct label ($Y = 1$ with probability $1-p$) leads to $a_i^{(1)} = 2(1-p)a_{i-1}^{(1)}$ and

$$\frac{a_i^{(1)}}{a_{i-1}^{(1)}} = 2(1-p), \quad \frac{b_i^{(1)}}{b_{i-1}^{(1)}} = \frac{1 - 2(1-p)(1-b_{i-1}^{(1)})}{b_{i-1}^{(1)}}.$$

A wrong label leads to

$$\frac{a_i^{(1)}}{a_{i-1}^{(1)}} = 2p, \quad \frac{b_i^{(1)}}{b_{i-1}^{(1)}} = \frac{1 - 2p(1-b_{i-1}^{(1)})}{b_{i-1}^{(1)}},$$

Therefore,

$$\mathbb{E}\left\{\ln\left(\frac{a_i^{(1)}}{a_{i-1}^{(1)}}\right)\right\} = (1-p)\ln\{2(1-p)\} + p\ln(2p) = \ln(2) - H(p) > 0, \quad \mathbb{E}\left\{\ln\left(\frac{b_i^{(1)}}{b_{i-1}^{(1)}}\right)\right\} < 0.$$

**Case 3, $\theta^* < X_i < \delta$.** The update rule for $a_i^{(1)}$ is exactly the same as Case 2, hence we have

$$\mathbb{E}\left\{\ln\left(\frac{a_i^{(1)}}{a_{i-1}^{(1)}}\right)\right\} = (1-p)\ln\{2(1-p)\} + p\ln(2p) = \ln(2) - H(p), \quad \mathbb{E}\left\{\ln\left(\frac{b_i^{(1)}}{b_{i-1}^{(1)}}\right)\right\} < 0.$$

For now, we assume that $2p < a_0^{(1)} < 1/2$; otherwise we can start the count of $N_1$ at the first time $t$ such that $2p < a_t^{(1)} < 1/2$, as explained later. Therefore, $\tau_{2k-1}, k = 1, 2, \ldots$ is the time that $a_t^{(1)}$ goes up and crosses $1/2$, while $\tau_{2k}, k = 1, 2, \ldots$ is the time that $a_t^{(1)}$ goes down and crosses $1/2$, and $T$ is the number of total cross times.

We note that $Z_i := \tau_i - \tau_{i-1}, i = 1, 2, \ldots$ are IID random variables with $\mathbb{E}Z_k \leq z$, where $z$ is a constant. To see it, we have

$$a_{\tau_{2k-1}}^{(1)} = a_{\tau_{2k-2}}^{(1)} \exp\left\{\sum_{i=0}^{\tau_{2k-1}-\tau_{2k-2}} \ln\left(\frac{a_{\tau_{2k-2}+i}^{(1)}}{a_{\tau_{2k-2}+i-1}^{(1)}}\right)\right\} \geq \exp\left(\sum_{i=1}^{\tau_{2k-1}-\tau_{2k-2}} V_i\right)/(2p),$$

$$b_{\tau_{2k}}^{(1)} = b_{\tau_{2k-1}}^{(1)} \exp\left\{\sum_{i=0}^{\tau_{2k}-\tau_{2k-1}} \ln\left(\frac{b_{\tau_{2k-1}+i}^{(1)}}{b_{\tau_{2k-1}+i-1}^{(1)}}\right)\right\} \leq 2(1-p)\exp\left(-\sum_{i=1}^{\tau_{2k}-\tau_{2k-1}} V_i\right), \quad (6)$$

where $V_i$'s are independent random variable with $\mathbb{E}V_i = \ln(2) - H(p) := v$. Moreover, $V_i$'s are uniformly bounded by a constant of $p$ solely, denoted by $B$. Therefore, $\ln(a_t^{(1)})$ is a random walk starting from (or above) $-\ln(2p)$ with a positive drift, which is expected to across $\ln(1/2)$ in a finite time by random walk theory (can be easily verified by applying Hoeffding's inequality). Similarly, $\ln(b_t^{(1)})$ is a random walk starting from (or below) $\ln(2(1-p))$ with a negative drift. We further define $S_l = \sum_{k=1}^{l} Z_k - kz$ and $S_0 = 0$. Clearly, $S_l$ is a super-martingale. Finally, optional stopping theorem yields that $\mathbb{E}S_T \leq S_0$, leading to

$$\mathbb{E}(T+1)z \geq \mathbb{E}\left\{\sum_{k=1}^{T+1}(\tau_k - \tau_{k-1})\right\} = \mathbb{E}(\tau_{T+1}) \geq n.$$

As a result, we have $\mathbb{E}(T) \geq \eta n$ for $\eta = 1/(2z)$.

Finally, we show that we can assume $2p < a_t^{(1)} < 1/2$. By Hoeffding's inequality, $a_t^{(1)}$ will across $1/2$ both up and down at least once within $n/2$ steps with probability at least $1 - e^{-C_4 n}$ with some constant $C_4$. Ever since that, we will have $2p \leq a_{\tau_i}^{(1)} < 1/2$ when $\text{sign}(a_{\tau_i}^{(1)}) = -1$ and $1/2 \leq a_{\tau_i}^{(1)} < 2(1-p)$ when $\text{sign}(a_{\tau_i}^{(1)}) = 1$, due to the update rule. We therefore conclude the proof. $\square$

## 3 Conclusion and Further Remarks

This work investigates the dynamics of PBA queries, revealing the intriguing pattern that they oscillate around the truth while steadily converging toward it. Building on this insight, we establish the exponential convergence rate of PBA, thereby bridging the long-standing gap between its theoretical guarantees and empirical performance. A natural direction for future research is to examine whether PBA still converges exponentially when the actual noise level $p$ exceeds the one assumed in the update rule, and, if not, to determine the resulting convergence rate. Another intriguing problem is the implementation of PBA, as it may be numerically challenge to exactly find the posterior median.

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

## A  Missing Proofs.

**Proof of Theorem 1**

*Proof.* We handle the case with $\theta^* \in (0, 1)$ first, and defer proof of the corner case to the end of this proof. For now, let $\delta$ be a constant such that $1 > \delta > \theta^* > 0$. We denote

$$M_i(\delta) := \frac{a_i^{(3)}(\delta)}{a_i^{(2)}(\delta)}. \tag{7}$$

Proposition 1 shows that for some constant $C > 0$ of $p$ only, we have

$$\mathbb{E} M_n \leq \frac{e^{-Cn}}{|\theta^* - \delta|}. \tag{8}$$

Since $a_i^{(j)} \in (0, 1)$ for $j = 1, 2, 3$ and all $i \geq 0$, we know that $M_n$ is positive, and Markov's inequality gives that

$$\mathbb{P}(a_n^{(3)} \geq \epsilon) \leq \mathbb{P}(M_n \geq \epsilon) \leq \frac{\mathbb{E} M_n}{\epsilon} \leq \frac{e^{-Cn}}{|\theta^* - \delta|\epsilon}.$$

When $0 < \delta < \theta^* < 1$, we can apply Proposition 1 after performing the transformation $x' = 1 - x$, which yields $\mathbb{P}(1 - a_n^{(3)} \geq \epsilon) \leq e^{-Cn}/(|\theta^* - \delta|\epsilon)$. Therefore,

$$\mathbb{P}(\min\{a_n^{(3)}, 1 - a_n^{(3)}\} \geq \epsilon) \leq e^{-Cn}/(|\theta^* - \delta|\epsilon). \tag{9}$$

When $\theta^* = 0$ or $\theta^* = 1$, we have $\mathbb{P}(\min\{a_n^{(3)}, 1 - a_n^{(3)}\} \geq \epsilon) = 0$ by definition (see, Eq. 7), therefore satisfying Eq. 9 as well.

Now, let $\delta_i = i/K, i = 0, \ldots, K$, where $K$ will be determined shortly. If $\min_i |\theta^* - \delta_i| < 1/\{2K(K+1)\}$, we can increase $K$ by 1, which ensures that $\min_i |\theta^* - \delta_i| \geq 1/\{2K(K+1)\}$. Clearly, there exists some $i^* \geq 1$ such that $\theta^* \in (\delta_{i^*-1}, \delta_{i^*})$. Evoking Eq. 9, we know that with probability at least $1 - 2K(K+1)e^{-Cn}/\epsilon$, we have

$$\mathbb{P}_n(X \in (\delta_{i^*-1}, \delta_{i^*})) \geq 1 - 2\epsilon,$$

implying that $|X_{n+1} - \theta^*| \leq 1/K$ for any $\epsilon < 1/4$. Therefore, for $K > 4$ and $\epsilon = 1/K$, we have

$$\mathbb{P}(|X_{n+1} - \theta^*| > 1/K) \leq 2K^2(K+1)e^{-Cn}.$$

Finally, taking $K = e^{Cn/4}$ yields

$$\mathbb{E}|X_{n+1} - \theta^*| \leq 1/K + \mathbb{P}(|X_{n+1} - \theta^*| > 1/K) \leq 3e^{-Cn/4}.$$

Regarding the corner case, we analyze with $\theta^* = 0$ as $\theta^* = 1$ can be handled with an analogous argument. In this case, Lemma 2 gives that

$$\mathbb{E}\left(a_n^{(3)}\right) \leq e^{-Cn}.$$

Let $\delta = \epsilon = 1/K$. Similar to the argument in the case of $\theta^* \in (0, 1)$, we have with probability at least $1 - e^{-Cn}/\epsilon$,

$$\mathbb{P}_n(X \in [0, \delta]) \geq 1 - \epsilon.$$

Choosing $K = e^{Cn/2}$ gives

$$\mathbb{E}|X_{n+1} - \theta^*| \leq 1/K + \mathbb{P}(|X_{n+1} - \theta^*| > 1/K) \leq 2e^{-Cn/2}.$$

We thus conclude the proof. $\square$

**Lemma 1.** *When $X_i \in (\theta^*, \delta)$, we have the following facts: (1) $q_1 + q_2 + q_3 = 1$. (2) $q_1, q_3 \in (0, 1/2)$. (3) $|q_1 - q_3| < q_2$. (4) For any $\epsilon < 1/2$, $q_2 - |q_1 - q_3| \geq \epsilon$ if and only if $q_1, q_3 \leq (1 - \epsilon)/2$.*

*Proof.* Fact (1) is by definition of $a_{i-1}^{(j)}$, $j = 1, 2, 3$. Their summation equals to $\mathbb{P}_{i-1}(X \leq 1) = 1$.

Fact (2) holds since $X_i \in (\theta^*, \delta)$; otherwise, if $q_1 \geq 1/2$ for example, we have $X_i \leq \theta^*$ since $P_{i-1}(X < \theta^*) = q_1 \geq 1/2$, which is a contradiction.

We prove Fact (3) by contradiction. If $|q_1 - q_3| \geq q_2$, then $q_1 \geq q_2 + q_3$ or $q_3 \geq q_1 + q_2$. However, $q_1 \geq q_2 + q_3$ with Fact (1) imply that $q_1 \geq 1/2$, which is a contradiction to Fact (2). Similarly, $q_3 \geq q_1 + q_2$ cannot hold as well.

Regarding (4), we use a similar argument as (3). Note that

$$q_2 - |q_1 - q_3| < \epsilon$$
$$\Longleftrightarrow q_1 > q_2 + q_3 - \epsilon \text{ or } q_3 > q_2 + q_1 - \epsilon$$
$$\Longleftrightarrow q_1 > (1 - \epsilon)/2 \text{ or } q_3 > (1 - \epsilon)/2.$$

We thus complete the proof. □

**Lemma 2.** *When $\theta^* = 0$ and $\delta < 1$, for some constant $C_1, C_2 > 0$ of $p$ only, we have*

$$\mathbb{P}(a_n^{(3)} \leq e^{-C_1 n}) \geq 1 - e^{-C_2 n},$$

*which implies $\mathbb{E}a_n^{(3)} \leq e^{-C_3 n}$ for some positive constant $C_3$.*

*Proof.* The spirit of this proof is the same as Proposition 1. Instead of studying the change of $M_i$, we can directly focus on $a_i^{(3)}$ when $\theta^* = 0$. Notably, when $\theta^* = 0$, there are only two potential locations of $X_i$.

**Case 1:** $X_i \in I_2$. A correct label ($Y_i = 1$ with probability $1 - p$) leads to $a_i^{(3)} = 2pa_{i-1}^{(3)}$, while a wrong label leads to $a_i^{(3)} = 2(1 - p)a_{i-1}^{(3)}$. As a result,

$$\mathbb{E}\left\{\ln\left(\frac{a_i^{(3)}}{a_{i-1}^{(3)}}\right)\right\} = (1 - p)\ln(2p) + p\ln(2(1 - p)) < 0.$$

**Case 2:** $X_i \in I_3$. Now, a correct label ($Y_i = 0$ with probability $1 - p$) leads to $a_i^{(3)} = 1 - 2(1 - p)a_{i-1}^{(3)}$, while a wrong label leads to $a_i^{(3)} = 1 - 2pa_{i-1}^{(3)}$. Therefore, by Jensen's inequality, we have

$$\mathbb{E}\left\{\ln\left(\frac{a_i^{(3)}}{a_{i-1}^{(3)}}\right)\right\} \leq \ln\left\{\mathbb{E}\left(\frac{a_i^{(3)}}{a_{i-1}^{(3)}}\right)\right\}$$
$$= \ln\left\{(1 - p)\frac{1 - 2(1 - p)a_{i-1}^{(3)}}{1 - a_{i-1}^{(3)}} + p\frac{1 - 2pa_{i-1}^{(3)}}{1 - a_{i-1}^{(3)}}\right\}$$
$$= \ln\left\{1 - \frac{a_{i-1}^{(3)}}{1 - a_{i-1}^{(3)}}(1 - 2p)^2\right\}$$
$$< 0.$$

With a similar argument as Proposition 1, we only have to show that Case 1 occurs sufficiently many times. Specifically, we define $\widetilde{\tau}_i = \inf_{t > \widetilde{\tau}_{i-1}} \text{sign}(a_t^{(3)} \neq \text{sign}(a_{t-1}^{(3)}), i = 1, 2, \ldots, \widetilde{\tau}_0 = 0$, and $\widetilde{T} = \sup_{i \geq 0}\{\widetilde{\tau}_i \leq n\}$. We show $\mathbb{E}\widetilde{T} \geq \eta n$ for some constant $\eta$ by tracking $a_i^{(2)}$. When $X_i \in I_3$, a correct label ($Y_i = 0$ with probability $1 - p$) leads to $a_i^{(2)} = 2(1 - p)a_{i-1}^{(2)}$, while a wrong label leads to $a_i^{(2)} = 2pa_{i-1}^{(2)}$. Therefore, we have

$$\mathbb{E}\left\{\ln\left(\frac{a_i^{(2)}}{a_{i-1}^{(2)}}\right)\right\} = (1 - p)\ln(2(1 - p)) + p\ln(2p) = \ln(2) - H(p) > 0.$$

The rest of proof is akin to Proposition 2. We thus complete the proof. □

## B  Discussion on the Noise Level

Our results apply to general responses $Y$ with a noise level up to $p$. That is, $\mathbb{P}(Y = h_{\theta^*}(X)) \geq 1 - p$ and $\mathbb{P}(Y = 1 - h_{\theta^*}(X)) \leq p$. To see it, an intuitive explanation is that at each round $i$, a correct response will drive the PBA estimator to be closer to the truth $\theta^*$, while an incorrect response will push it away from the truth. As a result, a higher noise level corresponds to a harder learning problem, and we discuss the most difficult learning scenario $(\mathbb{P}(Y = 1 - h_{\theta^*}(X)) = p)$ in the main paper.

Technically, inspecting the proof of Proposition 1, we find that the expectation of $M_i/M_{i-1}$ is even smaller when the probability of incorrect label is smaller than $p$. Meanwhile, the crossing time $T$ is still guaranteed to be at the order of $O(n)$. Therefore, the probability of an ill-performed estimator remains an exponentially decaying rate.

## C  Extension to High Dimensional Data

In this section, we extend our results to high dimensional setting where $d \geq 2$.

**Setup.**  Consider the query $X \in [0,1]^d, d \geq 2$ and the label $Y \in \{0,1\}$. Similar to the setting when $d = 1$, let $p \in (0, 1/2)$ represent the noise level in the labels, $\mathbb{P}(Y = h(X)) = 1 - p$ and $\mathbb{P}(Y = 1 - h(X)) = p$, where $h$ is a classifier $h : [0,1]^d \to \{0,1\}$, which a learner wants to estimate. Recall that in one dimensional setting, we consider a hypothesis class $\mathcal{H} = \{h_\theta : \theta \in [0,1]\}$ and work with a threshold classifier $h_{\theta^*}(x) = \mathbb{1}_{x \geq \theta^*}$. This ordered, one-parameter structure enables a probabilistic bisection algorithm (PBA, see Section 2), yielding an estimator $\widehat{\theta}_n$ which converges to $\theta^*$ exponentially fast, i.e. $\mathbb{E}|\widehat{\theta}_n - \theta^*| \leq \mathcal{O}(e^{-n})$ (see Theorem 1).

For $d \geq 2$, the natural analogue of a "threshold" is a decision boundary, whose shape should be restricted by additional geometric assumptions, such as smoothness, to ensure identifiability and control the complexity of the hypothesis class. In this work, we adopt a standard assumption in the literature that the decision boundary is Hölder smooth (Castro & Nowak, 2007; 2008). In particular, we consider the hypothesis class $\mathcal{H} = \{h_g : g \in \Sigma(L, \alpha)\}$, where $\Sigma(L, \alpha)$ denotes $\alpha$-Hölder smooth with parameters $L$ (see Definition 1).

**Definition 1.**  A function $g : [0,1]^{d-1} \to \mathbb{R}$ is Hölder smooth if it has continuous partial derivatives up to order $k = \lfloor \alpha \rfloor$ and $\forall \boldsymbol{z}, \boldsymbol{x} \in [0,1]^{d-1} :, g(\boldsymbol{z}) - \mathrm{TP}_{\boldsymbol{x}}(\boldsymbol{z}) \leq L\|\boldsymbol{z} - \boldsymbol{x}\|^\alpha$, where $L, \alpha > 0$, and $\mathrm{TP}_{\boldsymbol{x}}(\cdot)$ denotes the order $k$ Taylor polynomial approximation of $g$ expanded around $\boldsymbol{x}$.

The classifier is $h_{g^*}(x) = \mathbb{1}_{x \in G^*}$, where $g^*$ is the decision boundary of $G^*$ and $G^* = \{(\widetilde{X}, x_d) \in [0,1]^{d-1} \times [0,1] : x_d \geq g^*(\widetilde{X})\}$. In the following, we use $h, g^*$, and $G^*$ interchangeably. The learner wants to construct an estimator $\widehat{g}_n$, or equivalently, a classifier $\widehat{G}_n = \{(\widetilde{X}, x_d) \in [0,1]^{d-1} \times [0,1] : x_d \geq \widehat{g}_n(\widetilde{X})\}$, with small expected $L_1$ error $\mathbb{E}\|\widehat{g}_n - g^*\|_1$.

**Theorem 2.**  *There exists an estimator $\widehat{g}_n$ such that $\mathbb{E}\|\widehat{g}_n - g^*\|_1 \leq \mathcal{O}\left(\left(\frac{\log n}{n}\right)^{\frac{\alpha}{d-1}}\right)$.*

In the proof, we explicitly construct $\widehat{g}_n$ by generalizing the PBA to $d \geq 2$. At a high level, we recursively partition the $(d-1)$-dimensional base domain into dyadic cells and on each vertical lines, we deploy a one-dimensional PBA to localize the decision boundary within each cell. By combining these local estimates across the cells, we obtain a piecewise approximation of the boundary. The Hölder regularity of $g^*$ governs both the approximation error within each cell and the number of cells required at a given resolution, allowing a sample allocation that achieves the convergence rate in Theorem 2. Moreover, the matching information-theoretic lower bound of Castro & Nowak (2008) for learning Hölder-smooth decision boundaries implies that no estimator can achieve $L_1$ error smaller than a constant multiple of $n^{-\frac{\alpha}{(d-1)}}$ (up to logarithmic factors). Therefore, the upper bound in Theorem 2 is nearly minimax optimal.

**Special Case of $\alpha = \infty$.** A linear decision boundary is arbitrarily smooth, corresponding to the special case of $\alpha = \infty$. Theorem 2 implies that learning such a function using a PBA-based algorithm is faster than any polynomial rate. In fact, Theorem 3 below shows that one can still achieve an optimal exponential rate by leveraging PBA.

**Theorem 3.** *When the true boundary $g^*$ is linear, corresponding to the case $\alpha = \infty$, there exists an estimator $\widehat{g}_n$ satisfying $\mathbb{E}\|\widehat{g}_n - g^*\|_1 \leq C_1 \exp(-cn)$, where $C_1 > 0$ depending only on $d$ and $c > 0$ depending only on $d$ and the noise level $p$.*

**Proof of Theorem 2.**

*Proof.* **Estimator: constructing $\widehat{g}_n(\cdot)$ by grid–lines–interpolate.**

Pick an integer $M \geq 2$ and set $h = 1/M$. For each multi-index $\widetilde{\ell} \in \{0, \ldots, M\}^{d-1}$ let the base-grid node be $\widetilde{\boldsymbol{x}}_{\widetilde{\ell}} := M^{-1}\widetilde{\ell} \in [0,1]^{d-1}$. Along the vertical line $L_{\widetilde{\ell}} = \{(\widetilde{\boldsymbol{x}}_{\widetilde{\ell}}, x_d) : x_d \in [0,1]\}$, we collect $N$ samples and run a 1-d threshold estimator (using PBA as described in Section 2) to obtain $\widehat{g}(\widetilde{\boldsymbol{x}}_{\widetilde{\ell}})$ as an estimate $g^*(\widetilde{\boldsymbol{x}}_{\widetilde{\ell}})$. This yields a total of $N(M+1)^{d-1}$ samples, where the total number of samples $n$ satisfying $n \geq N(M+1)^{d-1}$. We then interpolate the estimates of $g^*$ at these points to construct a final estimates of the decision boundary.

In particular, we begin by dividing $[0,1]^{d-1}$ in to cells. Without of generality, we assume that $\alpha > 1$ ($\alpha = 1$ can be handled in similar way) and $\frac{M}{\lfloor \alpha \rfloor}$ is an integer (since this can always be achieved by the proper choice of $M$). For the ease of notation, let $r := \lfloor \alpha \rfloor \in \{1, 2, \ldots\}$, and let the cell index $\widetilde{q} = (\widetilde{q}_1, \ldots, \widetilde{q}_{d-1}) \in \{0, \ldots, \frac{M}{r} - 1\}^{d-1}$ define the axis-aligned cell $I_{\widetilde{q}} = \prod_{i=1}^{d-1} [\frac{r\widetilde{q}_i}{M}, \frac{r(\widetilde{q}_i+1)}{M}]$. In this way, the $(r+1)^{d-1}$ lattice nodes inside $I_{\widetilde{q}}$ have multi-indices $\widetilde{\ell} = (\ell_1, \ldots, \ell_{d-1}), \ell_i \in \{r\widetilde{q}_i, \ r\widetilde{q}_i + 1, \ \ldots, \ r\widetilde{q}_i + r\}$, and coordinates $\widetilde{\boldsymbol{x}}_{\widetilde{\ell}} := M^{-1}\widetilde{\ell}$. For bookkeeping in coordinate $i$, set the node locations $z_{i,j} := \frac{r\widetilde{q}_i + j}{M}, j = 0, 1, \ldots, r$, and the local index of $\ell_i$ within its cell $m_i := \ell_i - r\widetilde{q}_i \in \{0, 1, \ldots, r\}$.

Given these notations, we construct $\widehat{g}_n(\cdot)$ by the piecewise polynomial, shown as follows.

$$\widehat{g}_n(\widetilde{\boldsymbol{x}}) = \sum_{\widetilde{q}} \widehat{L}_{\widetilde{q}}(\widetilde{\boldsymbol{x}}) \, \mathbf{1}\{\widetilde{\boldsymbol{x}} \in I_{\widetilde{q}}\}, \tag{10}$$

where $\widehat{L}_{\widetilde{q}}(\widetilde{\boldsymbol{x}}) = \sum_{\widetilde{\ell}: \widetilde{\boldsymbol{x}}_{\widetilde{\ell}} \in I_{\widetilde{q}}} \widehat{g}(\widetilde{\boldsymbol{x}}_{\widetilde{\ell}}) \, Q_{\widetilde{q}, \widetilde{\ell}}(\widetilde{\boldsymbol{x}})$, and $Q_{\widetilde{q}, \widetilde{\ell}}(\widetilde{\boldsymbol{x}})$ is the multidimensional tensor-product basis on the cell. In particular,

$$Q_{\widetilde{q}, \widetilde{\ell}}(\widetilde{\boldsymbol{x}}) := \prod_{i=1}^{d-1} L_{i, \widetilde{q}_i, \ell_i}(\widetilde{\boldsymbol{x}}_i) = \prod_{i=1}^{d-1} \prod_{\substack{j=0 \\ j \neq m_i}}^{r} \frac{\widetilde{\boldsymbol{x}}_i - \frac{r\widetilde{q}_i + j}{M}}{\frac{\ell_i}{M} - \frac{r\widetilde{q}_i + j}{M}},$$

where $L_{i, \widetilde{q}_i, \ell_i}(t) := \prod_{\substack{j=0 \\ j \neq m_i}}^{r} \frac{t - z_{i,j}}{z_{i, m_i} - z_{i,j}}$. $\widehat{g}_n(\cdot)$ defines a classification rule $\widehat{G}_n$.

By Equation 10, we have the follows.

$$\mathcal{O}(\|\widehat{g}_n - g^*\|_1) = \mathcal{O}\Big(\sum_{\widetilde{q}} \|(\widehat{L}_{\widetilde{q}} - g^*)\mathbf{1}\{\widetilde{\boldsymbol{x}} \in I_{\widetilde{q}}\}\|_{L^1([0,1]^{d-1})}\Big)$$

$$= \mathcal{O}\Big(\sum_{\widetilde{q}} \|(L_{\widetilde{q}} - g^*)\mathbf{1}\{\widetilde{\boldsymbol{x}} \in I_{\widetilde{q}}\} + (\widehat{L}_{\widetilde{q}} - L_{\widetilde{q}})\mathbf{1}\{\widetilde{\boldsymbol{x}} \in I_{\widetilde{q}}\}\|_{L^1([0,1]^{d-1})}\Big)$$

$$= \mathcal{O}\Big(\sum_{\widetilde{q}} \|(L_{\widetilde{q}} - g^*)\mathbf{1}\{\widetilde{\boldsymbol{x}} \in I_{\widetilde{q}}\}\|_{L^1([0,1]^{d-1})} + \|(\widehat{L}_{\widetilde{q}} - L_{\widetilde{q}})\mathbf{1}\{\widetilde{\boldsymbol{x}} \in I_{\widetilde{q}}\}\|_{L^1([0,1]^{d-1})}\Big),$$

where $L_{\widetilde{q}}(\widetilde{\boldsymbol{x}}) = \sum_{\widetilde{\ell}: \widetilde{\boldsymbol{x}}_{\widetilde{\ell}} \in I_{\widetilde{q}}} g^*(\widetilde{\boldsymbol{x}}_{\widetilde{\ell}}) \, Q_{\widetilde{q}, \widetilde{\ell}}(\widetilde{\boldsymbol{x}})$ is the Clairvoyant version of $\widehat{L}_{\widetilde{q}}$.

Note that

$$\|(L_{\widetilde{q}} - g^*)\mathbf{1}\{\widetilde{\boldsymbol{x}} \in I_{\widetilde{q}}\}\|_{L^1([0,1]^{d-1})} = \int_{I_{\widetilde{q}}} |L_{\widetilde{q}}(\widetilde{\boldsymbol{x}}) - g^*(\widetilde{\boldsymbol{x}})|d\widetilde{\boldsymbol{x}} = \mathcal{O}\Big(\int_{I_{\widetilde{q}}} M^{-\alpha} d\widetilde{\boldsymbol{x}}\Big), \tag{11}$$

by using Lemma 3 and resulting in $\mathcal{O}(M^{-\alpha}M^{-(d-1)})$. Moreover, by conditioning on the good event where $|\widehat{g}(\widetilde{\boldsymbol{x}}_{\widetilde{l}}) - g^*(\widetilde{\boldsymbol{x}}_{\widetilde{l}})| \leq \epsilon_N$, we have

$$\|(\widehat{L}_{\widetilde{q}} - L_{\widetilde{q}})\mathbf{1}\{\widetilde{\boldsymbol{x}} \in I_{\widetilde{q}}\}\|_{L^1([0,1]^{d-1})} = \sum_{\widetilde{l}:\widetilde{\boldsymbol{x}}_{\widetilde{l}} \in I_{\widetilde{q}}} |\widehat{g}(\widetilde{\boldsymbol{x}}_{\widetilde{l}}) - g^*(\widetilde{\boldsymbol{x}}_{\widetilde{l}})| \|Q_{\widetilde{q},\widetilde{l}}\|_{L^1([0,1]^{d-1})} \quad (12)$$

$$\leq \sum_{\widetilde{l}:\widetilde{\boldsymbol{x}}_{\widetilde{l}} \in I_{\widetilde{q}}} \epsilon_N \left( \int_{I_{\widetilde{q}}} Q_{\widetilde{q},\widetilde{l}}(\widetilde{x}) d\mu \widetilde{\boldsymbol{x}} \right) \quad (13)$$

$$\leq \sum_{\widetilde{l}:\widetilde{\boldsymbol{x}}_{\widetilde{l}} \in I_{\widetilde{q}}} \epsilon_N \left( \int_{I_{\widetilde{q}}} r^{(d-1)r} d\mu \widetilde{\boldsymbol{x}} \right) \quad (14)$$

$$= \mathcal{O}\left( \epsilon_N M^{-(d-1)} \right). \quad (15)$$

Note that $\mu$ is a Lebesgue measure of $\widetilde{\boldsymbol{x}}$ which is uniform on $[0,1]^{d-1}$. By Equation 11 and 12, we have

$$\|\widehat{g}_n - g^*\|_1 \leq \mathcal{O}\Big(\sum_{\widetilde{q}} \|(L_{\widetilde{q}} - g^*)\mathbf{1}\{\widetilde{\boldsymbol{x}} \in I_{\widetilde{q}}\}\|_{L^1([0,1]^{d-1})} + \|(\widehat{L}_{\widetilde{q}} - L_{\widetilde{q}})\mathbf{1}\{\widetilde{\boldsymbol{x}} \in I_{\widetilde{q}}\}\|_{L^1([0,1]^{d-1})}\Big)$$

$$\leq \mathcal{O}\left( M^{d-1}(M^{-\alpha}M^{-(d-1)} + \epsilon_N M^{-(d-1)}) \right)$$

$$= \mathcal{O}\left( M^{-\alpha} + \epsilon_N \right).$$

According to Theorem 1, we know $\mathbb{P}(|\widehat{g}(\widetilde{\boldsymbol{x}}_{\widetilde{l}}) - g^*(\widetilde{\boldsymbol{x}}_{\widetilde{l}})| > \epsilon_N) \leq \frac{3}{\epsilon_N}\exp(-CN)$. Therefore, we choose $N = \lceil K\log n \rceil$, where $K > \frac{2\alpha}{C(d-1)}$, $M = \lfloor \left(\frac{n}{K\log n}\right)^{1/(d-1)} \rfloor$ and $\epsilon_N = \sqrt{3}\,e^{-cN/2}$, leading to

$$\mathbb{E}\|\widehat{g}_n - g^*\|_1 \leq \mathcal{O}\left( M^{-\alpha} + \epsilon_N \right) + \frac{3}{\epsilon_N}\exp(-CN) = \mathcal{O}\left( \left(\frac{\log n}{n}\right)^{\frac{\alpha}{d-1}} \right).$$

$\square$

**Lemma 3.** $\sup_{g^* \in \Sigma(L,\alpha)} \max_{\widetilde{\boldsymbol{x}} \in I_{\widetilde{q}}} |L_{\widetilde{q}}(\widetilde{\boldsymbol{x}}) - g^*(\widetilde{\boldsymbol{x}})| = \mathcal{O}(M^{-\alpha})$.

*Proof.* Let $\widetilde{\boldsymbol{x}} \in I_{\widetilde{q}}$ and $g \in \Sigma(L,\alpha)$, we have the follows.

$$\left| L_{\widetilde{q}}(\widetilde{\boldsymbol{x}}) - g^*(\widetilde{\boldsymbol{x}}) \right| = \left| L_{\widetilde{q}}(\widetilde{\boldsymbol{x}}) - \mathrm{TP}_{\widetilde{q}rM^{-1}}(\widetilde{\boldsymbol{x}}) - g^*(\widetilde{\boldsymbol{x}}) + \mathrm{TP}_{\widetilde{q}rM^{-1}}(\widetilde{\boldsymbol{x}}) \right|$$

$$\leq \left| L_{\widetilde{q}}(\widetilde{\boldsymbol{x}}) - \mathrm{TP}_{\widetilde{q}rM^{-1}}(\widetilde{\boldsymbol{x}}) \right| + \left| g^*(\widetilde{\boldsymbol{x}}) - \mathrm{TP}_{\widetilde{q}rM^{-1}}(\widetilde{\boldsymbol{x}}) \right|$$

$$\leq \left| L_{\widetilde{q}}(\widetilde{\boldsymbol{x}}) - \mathrm{TP}_{\widetilde{q}rM^{-1}}(\widetilde{\boldsymbol{x}}) \right| + L\left\| \widetilde{\boldsymbol{x}} - \widetilde{q}\,rM^{-1} \right\|^{\alpha}$$

$$\leq \left| L_{\widetilde{q}}(\widetilde{\boldsymbol{x}}) - \mathrm{TP}_{\widetilde{q}rM^{-1}}(\widetilde{\boldsymbol{x}}) \right| + \mathcal{O}(M^{-\alpha}).$$

Note that the tensor–polynomial approximation space contains the space of degree $r$ polynomials. Therefore we can write $L_{\widetilde{q}}(\widetilde{\boldsymbol{x}})$ as a tensor–product polynomial. Therefore, we have

$$\left| L_{\widetilde{q}}(\widetilde{\boldsymbol{x}}) - g^*(\widetilde{\boldsymbol{x}}) \right| \leq \left| \sum_{\widetilde{l}:\widetilde{\boldsymbol{x}}_{\widetilde{l}} \in I_{\widetilde{q}}} g^*(\widetilde{\boldsymbol{x}}_{\widetilde{l}}) Q_{\widetilde{q},\widetilde{l}}(\widetilde{\boldsymbol{x}}) - \mathrm{TP}_{\widetilde{q}rM^{-1}}(\widetilde{\boldsymbol{x}}) \right| + \mathcal{O}(M^{-\alpha})$$

$$= \left| \sum_{\widetilde{l}:\widetilde{\boldsymbol{x}}_{\widetilde{l}} \in I_{\widetilde{q}}} \left( g^*(\widetilde{\boldsymbol{x}}_{\widetilde{l}}) - \mathrm{TP}_{\widetilde{q}rM^{-1}}(\widetilde{\boldsymbol{x}}_{\widetilde{l}}) \right) Q_{\widetilde{q},\widetilde{l}}(\widetilde{\boldsymbol{x}}) \right| + \mathcal{O}(M^{-\alpha})$$

$$\leq \sum_{\widetilde{l}:\widetilde{\boldsymbol{x}}_{\widetilde{l}} \in I_{\widetilde{q}}} \left| g^*(\widetilde{\boldsymbol{x}}_{\widetilde{l}}) - \mathrm{TP}_{\widetilde{q}rM^{-1}}(\widetilde{\boldsymbol{x}}_{\widetilde{l}}) \right| \left| Q_{\widetilde{q},\widetilde{l}}(\widetilde{\boldsymbol{x}}) \right| + \mathcal{O}(M^{-\alpha})$$

$$\leq \sum_{\widetilde{l}:\widetilde{\boldsymbol{x}}_{\widetilde{l}}\in I_{\widetilde{q}}} L \left\| \widetilde{\boldsymbol{x}} - \widetilde{q}\,rM^{-1} \right\|^{\alpha} \left| Q_{\widetilde{q},\widetilde{l}}(\widetilde{\boldsymbol{x}}) \right| + \mathcal{O}(M^{-\alpha})$$

$$\leq \sum_{\widetilde{l}:\widetilde{\boldsymbol{x}}_{\widetilde{l}}\in I_{\widetilde{q}}} L \left\| \widetilde{\boldsymbol{x}} - \widetilde{q}\,rM^{-1} \right\|^{\alpha} r^{(d-1)r} + \mathcal{O}(M^{-\alpha})$$

$$\leq \sum_{\widetilde{l}:\widetilde{\boldsymbol{x}}_{\widetilde{l}}\in I_{\widetilde{q}}} \mathcal{O}(M^{-\alpha}) + \mathcal{O}(M^{-\alpha}) = r^{d-1}\mathcal{O}(M^{-\alpha}) + \mathcal{O}(M^{-\alpha}) = \mathcal{O}(M^{-\alpha}).$$

$$\square$$

**Proof of Theorem 3**

*Proof.* For a linear boundary, we denote $g^*(\widetilde{\boldsymbol{x}}) = a_*^\top \widetilde{\boldsymbol{x}} + b_*, \widetilde{\boldsymbol{x}} \in [0,1]^{d-1}$. Similar to the previous setting, the labels satisfy $\mathbb{P}\big(Y = h_{g^*}(X)\big) = 1 - p$ and $\mathbb{P}\big(Y = 1 - h_{g^*}(X)\big) = p$ with $p \in (0, \frac{1}{2})$. We show that there exist an estimator that achieves exponential $L_1$ error decay.

We first pick $m \geq d$ anchor points $\widetilde{\boldsymbol{x}}_1, \ldots, \widetilde{\boldsymbol{x}}_m \in [0,1]^{d-1}$ in general position so that the augmented design

$$Z = \begin{bmatrix} \widetilde{\boldsymbol{x}}_1^\top & 1 \\ \vdots & \vdots \\ \widetilde{\boldsymbol{x}}_m^\top & 1 \end{bmatrix} \in \mathbb{R}^{m \times d}$$

satisfies $\mathrm{rank}(Z) = d$. For each fixed $\widetilde{\boldsymbol{x}}_j$, query along the vertical line $\{(\widetilde{\boldsymbol{x}}_j, t) : t \in [0,1]\}$ and run a PBA to estimate the one-dimensional threshold $t_j^* := g^*(\widetilde{\boldsymbol{x}}_j) = a_*^\top \widetilde{\boldsymbol{x}}_j + b_*$.

One concrete choice with $m = d$ is to take $\widetilde{\boldsymbol{x}}_1 = \boldsymbol{0}, \widetilde{\boldsymbol{x}}_{k+1} = e_k$ for $k = 1, \ldots, d-1$, where $e_k$ denotes the $k$-th standard basis vector in $\mathbb{R}^{d-1}$. Then the augmented design matrix $Z$ is

$$Z = \begin{bmatrix} 0 & \cdots & 0 & 1 \\ 1 & 0 & \cdots & 1 \\ 0 & 1 & \cdots & 1 \\ \vdots & \vdots & \ddots & \vdots \\ 0 & 0 & \cdots & 1 \end{bmatrix} \in \mathbb{R}^{d \times d}.$$

Let $\widehat{t}_j$ be the PBA estimate after $n_j$ queries on line $j$, and set $\widehat{t} = (\widehat{t}_1, \ldots, \widehat{t}_d)^\top$. Estimate $\theta_* := (a_*, b_*) \in \mathbb{R}^d$ by least squares: $\widehat{\theta}_n := \arg\min_{\theta \in \mathbb{R}^d} \|\widehat{t} - Z\theta\|_2^2 = (Z^\top Z)^{-1} Z^\top \widehat{t}$, where $\widehat{t} = \widehat{g}(\widetilde{\boldsymbol{x}}) := \widehat{a}^\top \widetilde{\boldsymbol{x}} + \widehat{b}$. Let the total sample be $n = \sum_{j=1}^d n_j$. Let $\varepsilon := (\varepsilon_1, \ldots, \varepsilon_d)^\top$ with $\varepsilon_j = \widehat{t}_j - t_j^*$. Then we can express

$$\widehat{\theta}_n - \theta_* = (Z^\top Z)^{-1} Z^\top \varepsilon,$$

and $\|\widehat{\theta}_n - \theta_*\|_2 \leq \frac{1}{\sigma_{\min}(Z)} \|\varepsilon\|_2$, where $\sigma_{\min}(Z) > 0$ is the smallest singular value of $Z$. Let $\Delta a := \widehat{a} - a_*$ and $\Delta b := \widehat{b} - b_*$. Then we have

$$\mathbb{E}\|\widehat{g}_n - g^*\|_1 := \mathbb{E}\int_{[0,1]^{d-1}} \left| \Delta a^\top u + \Delta b \right| du \leq |\Delta b| + \tfrac{1}{2}\sum_{k=1}^{d-1} |\Delta a_k| \leq \left(1 + \tfrac{\sqrt{d-1}}{2}\right)\|\widehat{\theta} - \theta_*\|_2$$

$$\leq \frac{\left(1 + \tfrac{\sqrt{d-1}}{2}\right)}{\sigma_{\min}(Z)} \mathbb{E}\|\varepsilon\|_2 = \frac{\left(1 + \tfrac{\sqrt{d-1}}{2}\right)}{\sigma_{\min}(Z)} \mathbb{E}\|\widehat{t} - t^*\|_2 \leq \frac{\left(1 + \tfrac{\sqrt{d-1}}{2}\right)}{\sigma_{\min}(Z)} \sqrt{\sum_{j=1}^d \big(\mathbb{E}|\varepsilon_j|\big)^2}$$

$$\leq 3\frac{\left(1 + \tfrac{\sqrt{d-1}}{2}\right)}{\sigma_{\min}(Z)} \sqrt{d} \max_j \exp(-Cn_j).$$

We have the last equation by Theorem 1, which shows $\mathbb{E}|\widehat{t}_j - t_j^*| \leq 3\exp(-Cn_j), \forall j$, where $C$ is a constant of $p$ only. By taking $n_j = \frac{n}{d}$, we show that $\mathbb{E}\|\widehat{g} - g^*\|_1 \leq 3\frac{\left(1 + \tfrac{\sqrt{d-1}}{2}\right)}{\sigma_{\min}(Z)} \sqrt{d}\exp(-C\frac{n}{d})$.

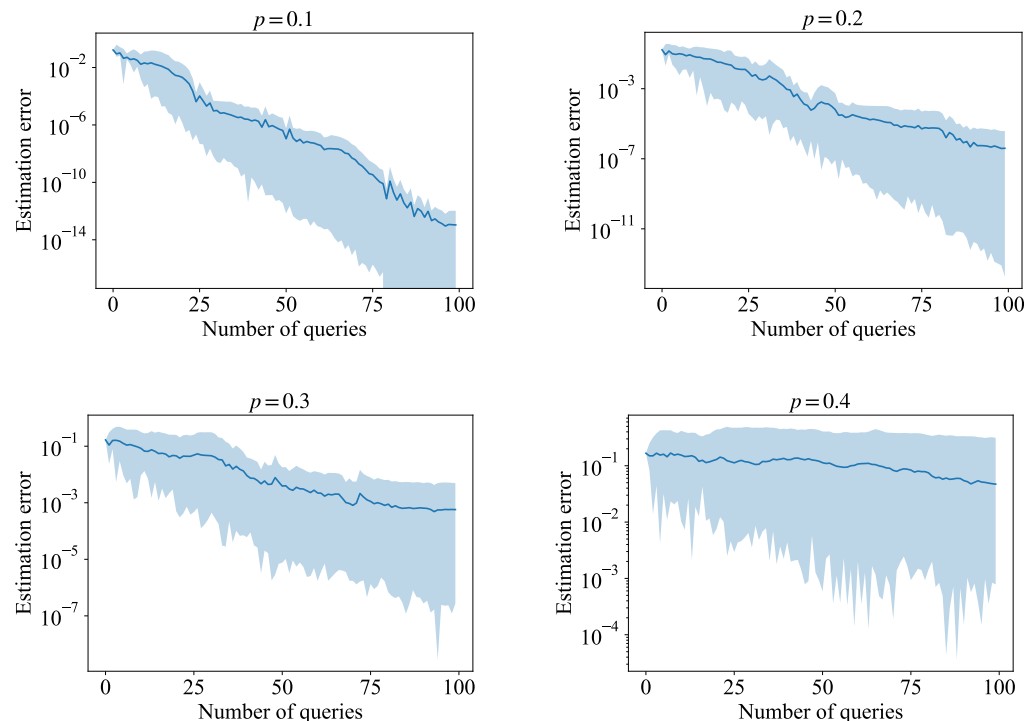

Figure 1: Estimator error rate of PBA estimator with respect to the query size $n$, under various noise level $p$.

Since the matrix $Z$ has full column rank and $\sigma_{\min}(Z)$ is bounded below by a positive constant depending only on $d$, we can write $\mathbb{E}\|\widehat{g} - g^*\|_1 \leq C_1 \exp(-cn)$, where $C_1 > 0$ depends only on $d$ and $c > 0$ depends only on $p$ and $d$. $\qquad\square$

## D  EXPERIMENTS

In this section, we conducted simulation experiments to corroborate our theoretical findings. WLOG, we choose $\theta^* = 1/3$ and vary the noisy level $p$ from a list of values $0.1, 0.2, 0.3, 0.4$. We report the average estimation error of the PBA estimator with respect to the query size $n$ on 20 replicated experiments. The results are shown in Figure 1. The maximum query size is 100 because the convergence rate is exponentially fast and the calculation of estimation error will encounter numerical issues, as seen in Figure 1.

Figure 1 clearly displays an exponential decay of the estimation error by PBA (a linear trend in the log-plot), aligned with our Theorem 1. In addition, a larger noise level $p$ results in a significantly smaller constant in the exponent of the convergence rate, leading to a slower convergence.

## E  THE USE OF LARGE LANGUAGE MODELS STATEMENT

Large language models were used solely as a writing aid. Their use was limited to minor language editing, such as correcting grammar, improving clarity, and polishing the phrasing, without altering the substantive content or analysis of the article.

