# OpenReview forum: "Probabilistic Bisection Algorithm Provably Achieves Exponential Convergence"
_ICLR.cc/2026/Conference — Submitted to ICLR 2026_

### Official Review · Reviewer_bKe2 · 2025-10-27

**Soundness:** 4
**Presentation:** 3
**Contribution:** 2
**Rating:** 4
**Confidence:** 4

**Summary:**

The paper analyzed the convergence rate of the Probabilistic Bisection Algorithm that computes the root of a function (framed as a binary classification problem). The algorithm is a natural heuristic that shares the same idea as binary search. In the binary classification formulated as used in the paper, we are given a function $1{x\geq \theta}$ where $\theta \in (0,1)$, and we want to know the value of $\theta$. At every time, we could query whether a point $x\geq \theta$; however, the answer could potentially be flipped with probability $p<1/2$. The PBA algorithm maintains a distribution over the support $[0,1]$: at round $r$, the algorithm samples a value of $x_{r}$ to query, and it gets the answer whether $x_{r} \geq \theta$ (with possible flipping noise). If the answer says $x_{r} \geq \theta$, we update the support by reweighing $f_r(x)=2 \cdot (1-p) \cdot f_{r-1}(x)$ for $x\leq x_{r}$ and $f_r(x)=2 \cdot p \cdot f_{r-1}(x)$ for $x\geq x_{r}$. This is similar to the ``binary search’’ process in the sense that since we’ve overshot, we want the next query to favor a search on the left side, and vice versa.

There have been various versions of analysis to provide theoretical justifications for the practical performances of the algorithm. The main contribution of this paper is a more ``generic’’ analysis for the convergent rate of the PBA algorithm, where the parameter $\theta$ and the final output $x_R$ are allowed to take values from the continuous domain of $[0,1]$. The paper showed that the distance between $x_R$ and $\theta$ decreases exponentially at a rate of $\exp(-\Omega(R))$ for $R$-round algorithms.

The central technique used in the paper is a new way to frame the dynamics of the PB algorithm. In particular, to control the upper tail of $x_R - \theta$, the analysis divides the query domains into $(0, \theta]$ vs. $(\theta, \delta)\cup [\delta, 1)$, and it uses the quantity of $M_r = \Pr(x \in [\delta, 1))/\Pr(x \in (\theta, \delta))$ to control the behavior of the PB algorithm. The paper showed that $M_r$ is a super-martingale, and the $M_{r+1} < M_r$ if the queries between $r$ and $r+1$ cross between $(0, \theta]$ and $(\theta, \delta)\cup [\delta, 1)$. Furthermore, the paper proved that the number of such crossings is at least $\Omega(R)$ for an $R$-round algorithm. Therefore, we can apply martingale concentration to get the desired result.

**Strengths:**

**This part contains the strength, the weakness, and my opinion.** I have mixed feelings about this paper. On one hand, this is a cute result of the convergence of the PB algorithm. The techniques are not involved, but they are nevertheless quite neat and not immediately straightforward. Therefore, I appreciate the value of the manuscript overall.

On the other hand, the setting seems very restrictive. As the paper has discussed, convergence bounds have been known for various versions of the random bisection algorithm, and the main contribution of this paper is the first bound for the continuous version of the algorithm. In other words, this is not the first paper that gives the conceptual message that explains the success of the algorithm in practice. This definitely limits the scope of the contribution.


The paper is well-written in general. However, there are a few issues I want to flag:
- The description (or definition) of the PB algorithm comes too late in the paper. I agree it’s quite an intuitive and simple algorithm; however, the paper spends quite a few passages discussing previous work without letting the readers know the algorithm, which makes it hard to understand.
- In your proof structure, the proof of proposition 1 was shown before proposition 2, but the proof of proposition 1 relies on proposition 2. The structure was not clear until I read the details on page 7 (which is also a bit too notation-heavy).
- The paper gave some intuitions of the proofs, but they are only stated as the passage goes. It would be good if the explanations were given earlier in the paper.

In summary, I believe that although the paper does contain some nice ideas, the significance of the results cannot push the paper to the most competitive tier.

**Weaknesses:**

See above.

**Questions:**

Most of the questions are embedded in the weakness comments. Some lower-order questions and comments (mostly about exposition):
- Line 218: The core idea behind the proof of Proposition 1 is to show that $\ln(M_i)$’s form a submartingale ... Here, do you mean supermartingale? Submartingales are the sequences that increase. This does not seem to affect correctness, though.
- It seems the dependency on p for the number of crossing steps in your Proposition 2 is linear. Is this true? If yes, maybe it’ll be better if you write them out.
- The entire analysis for high-dimensional data is deferred to the appendix. Can you explain what are the main ideas there?

---

> ### Author Response · Authors · 2025-11-17
>
> We sincerely thank the reviewer for their thoughtful and constructive feedback.
>
> **Scope of contribution.** We would like to clarify that our contribution is not merely an alternative explanation for the PBA’s success, but a foundational advance: to the best of our knowledge, this is the first and only direct convergence rate analysis of the original PBA under the setting where the ground truth $\theta^*$ is fixed but unknown.
>
> Historically, analyzing PBA's performance has been difficult due to the continuous nature of the query sequence, which makes tracking the estimation non-trivial. As a result, prior efforts either (a) adopted a discretized version of PBA, or (b) invoked a Bayesian framework where $\theta^*$ is modeled as a continuous random variable. However, these approaches are unable to directly characterize the convergence behavior of the original PBA given a fixed truth.
>
> Our work demonstrates that understanding the query behavior of PBA is both essential and powerful in tackling this problem. The techniques used in previous work do not apply in this setting, as they do not account for the relative location of queries to the fixed truth $\theta^*$, a central ingredient in our analysis. Specifically, Proposition 1 and 2  yield a sharp picture of  query behavior: PBA queries oscillate around the true root but gradually concentrate closer to it. This insight is key to deriving convergence rates for the original PBA.
>
> In addition, our analysis extends naturally to high-dimensional settings, where the root is replaced by a smooth decision boundary,  as outlined in Appendix C. This presents challenges that earlier techniques (e.g., those assuming a prior over $\theta^*$) cannot easily address.
>
> Moreover, our analysis supports query-specific noise models, enabling settings with non-IID noise. This is achieved by inspecting how each individual query contributes to estimation, and allows us to accommodate broader classes of practical applications. We believe this paves the way for more general analyses in active learning and related areas.
>
> **Response to Other Points**
>
> **Description of PBA.** Thank you for your suggestion. We have incorporated the excellent example in your summary as an informal and early-stage definition of PBA. We also note that a high-level description is already provided in the Introduction (Lines 43–45), but we’re happy to improve clarity.
>
> **Order of propositions and proof insights.**
> We appreciate the suggestion and have revised the manuscript to improve the logical flow and added more commentary to highlight the intuition behind key propositions. These discussions are added right after introducing propositions 1 and 2.
>
> **Line 218.** Yes, we mean supermartingale. Thank you for catching this typo.
>
> **Dependency on $p$.** No, $\eta$ does not necessarily depend linearly on $p$. From the proof of Proposition 2, the number of cross time is governed by two random walks with drift terms involving $\ln(2) - H(p)$, where $H(p)$ is the binary entropy function, yielding a non-linear relationship as captured in Eq. (6).
>
> **High-dim data.** In higher dimensions, our goal shifts to estimating a smooth decision boundary within $[0,1]^d$ (which degenerates to a single point when $d=1$). To handle this, we can partition the first $d-1$ dimensions $[0,1]^{d-1}$ into many equal-spaced grid points. Then, we apply 1-D PBA to estimate the decision boundary on the last dimension conditional on each grid point. Once we get accurate local estimations on the last dimension, we can interpolate them to the entire space to get the final estimation.
>
>
> We hope this clarification helps deliver the novelty, technical depth, and practical implications of our work, and we kindly request the reviewer to reconsider the merit of our contribution in this light.

---

### Official Review · Reviewer_gVAc · 2025-11-01

**Soundness:** 3
**Presentation:** 3
**Contribution:** 2
**Rating:** 4
**Confidence:** 3

**Summary:**

The paper studies the Probabilistic Bisection Algorithm (PBA), an extension of classical binary search that handles noisy feedback. The authors prove that PBA converges at a geometric rate, by showing that its queries oscillate around the true value while steadily approaching it.

**Strengths:**

1. The paper bridges the gap between PBA’s empirical performance and theoretical understanding by providing rigorous convergence guarantees.

2. It offers new insights into the behavior and dynamics of the Probabilistic Bisection Algorithm.

3. The paper is well-written and clearly organized, making the technical arguments easy to follow.

**Weaknesses:**

Although the paper studies a basic and fundamental problem in stochastic optimization and root-finding, it is unclear to me how relevant this topic is to the ICLR community. The work is purely mathematical, with only small-scale experiments serving as a proof of concept, which may limit its appeal to ICLR’s broader audience focused on learning algorithms and applications.

**Questions:**

N/A

---

> ### Author Response · Authors · 2025-11-17
>
> We sincerely thank the reviewer for their valuable time. We would like to respectfully clarify why we believe our work is relevant and potentially impactful to the ICLR community.
>
> **Relevance to ICLR**
>
> ICLR has a Learning Theory track and a strong history of welcoming purely theoretical work that advances our understanding of core learning principles. In fact, numerous theoretical papers have been published and some of them even selected as outstanding papers at ICLR in recent years [1–3]. We believe our submission aligns well with this tradition.
>
> **Importance of noisy stochastic optimization in machine learning.**
>
> As reviewer acknowledged, our paper addresses a fundamental problem in noisy stochastic optimization and root-finding, which are core components underpinning many modern machine learning algorithms. Methods such as hyperparameter tuning, quantile estimation, calibration, and certain classes of bandit problems can all be viewed as instances of noisy root-finding problems. Our work thus rigorously establishes the convergence guarantee of PBA, an efficient and commonly used algorithm to solve these problems, and provides insights over its empirical success.
>
> **Theoretical contribution and implications.**
>
> While our contribution is primarily theoretical, it addresses a long-standing problem in the literature by establishing the first convergence rate guarantees for PBA in the realistic setting where the root is fixed but unknown. We also develop novel tools to analyze query behavior. Those results are not only mathematically non-trivial but also inspiring for providing learning-theoretic guarantees for general online learning and active learning algorithms.
>
>
> We respectfully emphasize that ICLR has a strong precedent of publishing rigorous theoretical work that contributes to the foundations of learning systems. We believe our work falls within this category, and its foundational nature may benefit both theoretical and applied researchers in the community. Therefore, we kindly request the reviewer to reconsider the merit and contribution of our work.
>
> [1] Neyshabur B, Li Z, Bhojanapalli S, et al. Towards understanding the role of over-parametrization in generalization of neural networks. ICLR 2019.
>
> [2] Geerts F, Reutter J L. Expressiveness and approximation properties of graph neural networks. ICLR 2022.
>
> [3] Kolossov G, Montanari A, Tandon P. Towards a statistical theory of data selection under weak supervision. ICLR 2024.

---

### Official Review · Reviewer_KfBU · 2025-11-02

**Soundness:** 2
**Presentation:** 3
**Contribution:** 2
**Rating:** 2
**Confidence:** 3

**Summary:**

The paper analyzes the probabilistic bisection algorithm, which is an algorithm similar to bisection except designed for settings where responses are noisy. It analyzes the algorithm in the continuous setting, where the target we are searching for can be anywhere in a compact set.  It presents a new analysis that the article claims generalizes an assumption from a previously-published result by Waeber et al. 2013.

**Strengths:**

The paper considers an interesting question and presents a substantial theoretical analysis. If it weren't for the issue raised below under weaknesses, I would say that the paper is very interesting.  The proofs technique do seem to be new, even if the result doesn't seem stronger than the previous result (see below).

**Weaknesses:**

My main question about this paper is the difference between its results and those in Waeber et al. 2013.

The submitted paper states (lines 067-068) that

"By modeling the root as a random variable $X^\ast$ uniformly distributed on [0,1], they proved that $E|X^\ast-\hat{X}_n|$ decays geometrically... However this result hinges critically on the assumption that $X^\ast$ is a uniform random variable."

This assumption, that $X^\ast$ is uniformly distributed, seems to be the key distinction that the paper is drawing with Waeber et al. 2013.

But I looked at Waeber et al. 2013 and they *don't* seem to make an assumption that $X^\ast$ is uniform.  Page 2263 of the journal version of the papr states, "we assume it ($X^\ast$) is a realization of an absolutely continuous random variable with density $f_0$. The density $f_0$ has domain [0, 1] and is known." Later in the paper (page 2264) the paper states, "If we have no prior knowledge of $X^\ast$, then a natural choice of $f_0$ is the uniform U[0, 1] distribution, i.e., $f_0(y) = \{y \in [0, 1]\}$." But this appears to just be an example and not an assumption.

So the only assumption seems to be that it is absolutely continuous and has support in [0,1]. The result seems easy to extend to other supports, e.g., [a, b], just by defining a new $X^\ast$ that is a shifted and scaled version of the original.  The scaling will then show up in the bound on expected L1 error.

Without this key distinction, the results in the paper don't seem stronger than the ones in Waeber et al. 2023.

**Questions:**

Some other less important questions:

1. The text near the bottom of page 3 discusses a quantity $M_i = min(A_i, 1-A_i)$ from Waeber et al. 2013.  It states near equation 3 that "their proofs rely on the argument that $M_{i+1} / M_i \le \exp(-C)$." The text then goes on to say in boldface that "improvement is not always guaranteed".

A closer reading of Waeber et al. 2013 would show that actually the arguments there *don't* rely on the argument that $M_{i+1} / M_i \le \exp(-C)$, at least, not on every sample path.
Instead, $M_i$ is upper bounded by another quantity $S_i$ that is a geometric walk and satisfies the inequality in the sense that $\log S_i$ is a supermartingale, i.e., $E[ \log (S_{i+1} / S_i) ] \le -C$ for some constant C.

This argument is somewhat similar to the argument in the submitted paper --- page 5 shows that the $\log(M_i)$, as defined in the paper, is a submartingale.  If the paper is accepted, this should be clarified.


2. It would help to give a more clear high level summary of the differences in proof techniques between the submitted paper and Waeber et al. 2013.  Both seem to use a discretization of the space
and use a sub/super-martingale argument to show that the mass in incorrect parts of the discretization shrink exponentially fast. But the discretizations chosen are different as are the definitions of the sub/super-martingales.


3. When defining $a_i^{(j)}(delta)$ on line 177 of page 4, I recommend dropping the middle term, $P_i(X \in I_j)$, because it is confusing and it isn't equal to the right-hand side.  The middle term is equal to 1 for i=1, since $\theta^\ast$ is always in $I_1$, and is equal to 0 for i=2,3.  The tricky thing here is that we want to compute the posterior probability that $\theta^\ast \in I_j$ without knowing how $I_j$ was defined.

---

> ### Author Response · Authors · 2025-11-17
>
> We sincerely thank the reviewer for their thoughtful feedback. We would like to clarify our contribution and address the concerns regarding the relationship to Waeber et al. (2013), as well as the technical distinctions in our analysis.
>
> **Clarification of Contribution and Distinction from Waeber et al. (2013)**
>
> Our work does not generalize an assumption from Waeber et al. (2013), but instead presents a fundamentally different analysis of the convergence behavior of PBA.
>
> Specifically, Waeber et al. 2013 adopts a Bayesian framework that assumes $X^\star$ is a random variable with a continuous and bounded density on the entire support. On page 2265, they state "$X^\star$ is a random variable with density $f_0$." Their analysis focuses on the evolution of $A_n = P_n(X^\star \in [0, h))$, which is meaningful when $X^\star$ is a random variable. Notably, the assumption that $X^\star$ has the density $f_0$ results in a key step in their proof: $A_i$ increases and decreases with equal probability at each round. However, this trick does not work when $X^\star$ is a fixed value, and their results does not apply for a fixed truth.
>
> In contrast, our work considers the frequentist setting where $X^\star$ is fixed but unknown. This setting better reflects many real-world applications where the true value is not randomly drawn. Under this setting, the symmetry leveraged by Waeber et al. 2013 no longer holds, the behavior of the posterior distribution becomes dependent on the query sequence relative to the fixed $X^\star$.
>
> Consequently, our analysis must handle the lack of a symmetric random walk structure, requiring new techniques that analyze the dynamics of the query points directly. This results in a distinct proof strategy and new theoretical tools.
>
> **Response to Specific Points**
>
> **Weakness**
>
> As highlighted above, our key theoretical contribution lies in addressing the convergence of PBA when $X^*$ is fixed, a setting for which prior techniques, such as those in Waeber et al. 2013, are not applicable. Our results therefore provide new insights and tools on query behaviors that fill an important gap in the literature.
>
>
>
> **Q1 & Q2.**
>
> We appreciate your careful reading. It is correct that Waeber et al. 2013 rely on the observation that $E(\log(S_{i+1}/S_{i}) \mid S_i) \leq -C$ for some constant $C$.
>
> We intended to convey the message that this result is made possible by the Bayesian assumption that $X^\star$ has density $f_0$. Under this setting, $A_i$ has equal probability to go up or down at each round $i$, regardless of query $X_i$. This symmetry leads to a geometric random walk $S_i$ that upper bounds $A_i$, and the estimation is expected to be refined at every round.
>
> In our setting, where $X^\star$ is fixed, such symmetry does not exist. Instead, we find that the update of estimation is dependent of query location. We therefore perform a finer-grid analysis of query dynamics to address this issue, which has not been analyzed in previous work. In particular, we construct a different upper bound $M_i$, which (in expectation) does not monotonically decrease but still tightens the estimate when a query crosses the true value. We prove that such beneficial crossings occur frequently enough to ensure exponential convergence.
>
> In summary, while both works aim to bound the probability that PBA yields a bad estimator that are far away from the truth, our analysis explicitly handles the challenges introduced by a fixed unknown $X^\star$, a nontrivial setting that demands fundamentally different tools.
>
>
> **Q3.**
>
> We would like to clarify a potential misunderstanding. Again, in our paper, the truth $X^\star$ ($\theta^\star$ by our notation) is fixed. The distribution $P_n$ is an internal belief distribution maintained by the PBA algorithm. Therefore, when we compute $P_i(X \in I_j) = \int_{I_j} f_i(x)dx$ we refer to the probability mass under the belief distribution, not the probability that $X^\star \in I_j$. This is fundamentally different from Waeber et al. (2013), where $P_i(X^\star \in [0, h))$ has probabilistic meaning due to the assumption that $X^\star$ is a random variable.
>
>
> Based on our clarification, we thus kindly request the reviewer to re-evaluate the merit and contribution of our work in light of these distinctions. We are grateful for the reviewer’s time and thoughtful questions, and we welcome any further feedback.

---

### Author Response · Authors · 2025-11-29
**Revision Summary**

Dear Reviewers and Area Chair,

We sincerely appreciate your constructive feedback and the time you’ve dedicated to the review process. Below, we summarize the major revisions and clarifications we have made in response to your comments. These updates are highlighted in blue in the revised manuscript.

1. **Distinction from Existing Work.** In response to the comments from Reviewers KfBU and bKe2, we have clarified in the manuscript that our approach is fundamentally different from prior work such as Waeber et al. (2013). Specifically, we aim at understanding the convergence rate of the original PBA under a fixed, unknown truth. This requires a finer-grained analysis of the query behavior of PBA, an aspect that has not been explored before. Technically, previous analyses rely on an argument that the estimation quality improves on average at each round, which does not hold under our setting.

     Our analysis therefore (1) characterizes the query dynamics and provides theoretical justification for PBA’s empirical success; (2) naturally extends to high-dimensional settings; and (3) supports query-specific noise models, offering insights that may inspire broader analyses in active learning and related areas.

2. **Relevance to ICLR.** In response to Reviewer gVAc, we respectfully note that ICLR has a Learning Theory track and a strong history of welcoming purely theoretical work that advances our understanding of core learning principles. As reviewer acknowledged, our paper addresses a fundamental problem in noisy stochastic optimization and root-finding, which we believe fits well within this scope.

3. **Improved Readability.** Following the suggestions from Reviewer bKe2, we have revised the manuscript to improve clarity and presentation. Specifically, we now introduce PBA earlier in the introduction, better motivate our technical contributions, and reorganize the structure to enhance logical flow and readability.

We believe these revisions comprehensively address the reviewers’ concerns, and we kindly hope the Area Chair will take these updates into account during the final evaluation. Thank you again for your time and dedication to the review process!

Sincerely,

Authors of Submission 10091

---

### Meta-Review · Area_Chair_4WCj · 2025-12-28

**Summary:**

This paper proves a geometric convergence rate for the original PBA algorithm under a fixed unknown truth. While most reviewers find the analysis to be interesting, the reviewers are concerned with the novelty of the result, in particular compared to a previous work Waeber et al. (2013) and other works in discrete settings. The authors partially addressed this concern by pointing out that this work has a very different approach and can handle fixed unknown truth. However, this improvement over existing result still does not seem significant. As the reviewers pointed out, the conceptual message of this paper is not as strong as stated in the abstract. The authors should revise the paper to highlight why this particular setting of fixed unknown truth is of crucial importance and why previous techniques fundamentally cannot be adapted to solve this setting.

**Reviewer Concerns:**

The main concern is the overlap with prior work, which is partially addressed but still the work does not feel as significant as its abstract claims.

There is the minor concern that such a result is not suited for ICLR. I don't share this concern.

**Reviewer Scores:**

KfBU: given that the result is not exactly the same as Waeber et al, probably will +1
gVAc: unlikely to change
bKe2: unlikely to change

---

### Decision · Program_Chairs · 2026-01-26

Reject